# Gold Nanorods with Mesoporous Silica Shell: A Promising Platform for Cisplatin Delivery

**DOI:** 10.3390/mi14051031

**Published:** 2023-05-11

**Authors:** Jaime Quiñones, Fabiola Carolina Miranda-Castro, David Encinas-Basurto, Jaime Ibarra, Edgar Felipe Moran-Palacio, Luis Alberto Zamora-Alvarez, Mario Almada

**Affiliations:** 1Posgrado en Nanotecnología, Departamento de Física, Universidad de Sonora, Unidad Regional Centro, Hermosillo 83000, Mexico; a205208483@unison.mx (J.Q.); a206202841@unison.mx (F.C.M.-C.); 2Departamento de Física, Matemáticas e Ingeniería, Universidad de Sonora, Campus Navojoa, Navojoa 85880, Mexico; david.encinas@unison.mx (D.E.-B.); jaime.ibarra@unison.mx (J.I.); 3Departamento de Ciencias Químico-Biológicas y Agropecuarias, Universidad de Sonora, Lázaro Cárdenas 100, Colonia Francisco Villa, Navojoa 85880, Mexico; edgar.moran@unison.mx (E.F.M.-P.); luis.zamora@unison.mx (L.A.Z.-A.)

**Keywords:** gold nanorods, cisplatin, mesoporous shell, cancer

## Abstract

The versatile combination of metal nanoparticles with chemotherapy agents makes designing multifunctional drug delivery systems attractive. In this work, we reported cisplatin’s encapsulation and release profile using a mesoporous silica-coated gold nanorods system. Gold nanorods were synthesized by an acidic seed-mediated method in the presence of cetyltrimethylammonium bromide surfactant, and the silica-coated state was obtained by modified Stöber method. The silica shell was modified first with 3-aminopropyltriethoxysilane and then with succinic anhydride to obtain carboxylates groups to improve cisplatin encapsulation. Gold nanorods with an aspect ratio of 3.2 and silica shell thickness of 14.74 nm were obtained, and infrared spectroscopy and ζ potential studies corroborated surface modification with carboxylates groups. On the other hand, cisplatin was encapsulated under optimal conditions with an efficiency of ~58%, and it was released in a controlled manner over 96 h. Furthermore, acidic pH promoted a faster release of 72% cisplatin encapsulated compared to 51% in neutral pH.

## 1. Introduction

Cancer is a disease of global concern which caused about one-sixth of the total deaths in the world in 2020 [1]. Chemotherapy is one of the most important therapeutic strategies to overcome it, although its efficiency to eradicate all cancer types is limited [2]. Cisplatin is a metal-based chemotherapeutic drug commonly used for a wide range of solid cancers [3] however, it presents several side effects and poor solubility, and several cancers can develop drug resistance to it [4,5]. Nanomaterials offer a viable option to overcome these limitations by improving the biodistribution of cisplatin using several types of drug delivery systems [6]. 

Mesoporous silica nanoparticles (MSN) are of particular interest for drug delivery systems because they are easy to synthesize and functionalize. Furthermore, they present well-ordered internal pores (2–50 nm) with a considerably large pore volume (0.6–1 cm^3^/g) and surface area (700–1000 m^2^/g), and they can function as reservoirs for various molecules [7,8]. Cisplatin has been previously encapsulated in MSN; Rejeeth et al. [9] attached cisplatin to silica nanoparticles using aminopropyltriethoxy silane as a linker molecule, and the highest surface area incited the greatest response. Varache et al. [10] functionalized MSN with low-molecular-weight branched polyethyleneimine (PEI) to load cisplatin, and the system controlled drug release over 72 h with the absence of burst release. Additionally, MSN has been modified with carboxyl groups to improve loading capacity and cisplatin’s release profile by coordinating complex formation between carboxylates and platinum atoms by replacing chloride groups [11,12,13]. Gu et al. [11,12] synthesized carboxyl functionalized MSN to promote coordination of complex formation with cisplatin, leading to an increase in drug-loading efficiency and prolonged cisplatin release. Zhu et al. [13] synthesized magnetic nanoparticles initially covered with a silica layer functionalized with carboxyl groups for cisplatin encapsulation. The system showed great drug capacity, sustained release and higher cytotoxicities compared to free cisplatin in A549 and MCF-712 cell lines. Nejad and Urbassek [14] used dynamic simulation to determine the diffusion behavior of cisplatin in a water-filled silica nanopore. They showed that MSN pores can be used as cisplatin containers if the pore diameter is larger than around 2 nm; in this case, the drug will have a diffusion coefficient comparable to that in bulk water, allowing efficient cisplatin release from the container. The optimal pore size was surprisingly large, given that the van der Waals size of cisplatin is only around 6 Å (a factor of 3 or smaller). They suggested that this is probably due to adsorption processes at the inner pore walls which hinder the efficiency of drug diffusion out of the pore.

It has been demonstrated that hyperthermia potentiates the effect of cisplatin on tumor treatment by increasing cellular drug uptake and stimulating apoptosis [15,16]. It is well-known that gold nanoparticles (AuNP) can induce local hyperthermy by laser irradiation through a phenomenon known as surface plasmon resonance (SPR), which arises from coherent electron oscillation trough gold nanostructures’ surface [17,18]. In addition, AuNP have unique physicochemical properties that make them ideal for a wide range of biomedical applications, including drug delivery, imaging and therapy. They can also be used as contrast agents for imaging techniques such as computed tomography (CT) and optical imaging. The nanoparticles can be functionalized with targeting molecules to specifically bind to cells of interest, allowing for high-resolution imaging of specific tissues or organs. Popovtzer et al. [19] synthesized AuNPs to selectively target and visualize cancer cells using CT imaging as contrast agents, absorbing X-rays and generating a clear image of the tumor site, allowing the tumor to be recognized with high sensitivity and specificity.

Gold nanorods (AuNR) are an interesting AuNP type that show two different SPR modes, which is due to the oscillation of electrons across the transversal and longitudinal axes of surface AuNR. The last one is extremely sensitive to the aspect ratio, which provides a straightforward route to control its position, and thus, AuNR can be designed to absorb light in the near infrared (700–1000 nm), where absorption by tissues is minimal; this feature makes them very interesting for medical applications, biosensing, etc. [20,21]. AuNR can also be used in various diagnostic applications, for example, in surface-enhanced Raman spectroscopy (SERS), allowing for highly sensitive detection of analytes. Meyer and Murphy [22] coated AuNR with anisotropic silica using a layer-by-layer technique. The SERS signal of a molecule model was increased by more than five times compared to non-coated gold nanorods. The coated surface increased the local electromagnetic field surrounding the nanorods, amplifying the Raman scattering signal. They also reported that thicker coatings lower sensor sensitivity by interfering. Huang et al. [23] used folic-acid-functionalized gold nanorods as contrast agents for X-ray imaging and detected solid tumors in animal models. Locatelli et al. [24] synthesized a drug delivery system able to host two different therapeutic agents, Adriamycin and AuNR, simultaneously. They proposed photoacousting imaging to detect tumor regions, thus allowing diagnosis and therapy by using a single drug. The efficacy of the novel therapeutic agent was demonstrated both in vitro and in vivo.

Due to AuNR and MSN properties, the design and synthesis of a multifunctional drug delivery system to administer cisplatin and produce local hyperthermy is possible; this strategy has been reported previously using doxorubicin as a model drug [25,26]. Although MSN and AuNR have been reported on exhaustively in previous years as drug delivery systems and phototherapy agents, respectively, the combination of AuNR coated with MSN to encapsulate cisplatin has not been reported yet. In the present work the synthesis of carboxylate-modified mesoporous silica-coated gold nanorods to encapsulate cisplatin through the formation of coordinate complex is studied, and its characterization by SEM, FTIR, and ζ potential is shown; in addition, the cisplatin encapsulation and release profile are studied.

## 2. Materials and Methods

### 2.1. Materials

The following materials were used: cetylltrimethylammonium bromide (CTAB, 99%), sodium hydroxide (97%), ethyl alcohol (99.5%), tetra-ethyl-orthosilicate (TEOS, 99%), 3-aminopropyltriethoxysilane (APTES), ammonium nitrate (NH_4_NO_3_), succinic anhydride (SA), dimethylformamide (DMF), triethylamine ((N(CH_2_CH_3_)_3_), cis-diamminedichloroplatinum (II) (cis-[Pt(NH_3_)_2_Cl_2_]), dimethylsulfoxide (DMSO), ortho-phenylenediamine (OPDA), phosphate buffer (pH 6.8, 50 Mm), phosphate buffer saline (PBS, pH 7.4), disodium phosphate (Na_2_HPO_4_), monopotassium phosphate (KH_2_PO_4_), sodium chloride, potassium chloride, hydrogen tetrachloroaurate (III) (HAuCl_4_), silver nitrate (AgNO_3_), ascorbic acid and sodium tetrahydridoborate (NaBH_4_); water was obtained deionized milli-Q (resistivity 18.2 MΩ·-cm). All chemicals were used without further purification.

### 2.2. Synthesis of AuNR

The AuNR were synthesized by the seeds-mediated method under acidic conditions [25]. gold seeds were prepared by adding 0.5 mL of NaBH_4_ (10 mM) solution to 5 mL of an aqueous solution composed of CTAB (0.1 M) and HAuCl_4_ (0.48 mM). This solution was stirred vigorously for 2 min and then held at 33 °C for 30 min. The growth solution (100 mL) was prepared separately by mixing CTAB, HCl, HAuCl_4_, AgNO_3_ and ascorbic acid at final concentrations of 0.1 M, 0.08 M, 0.5 mM, 0.085 mM and 1 mM, respectively. The colorless growth solution was kept at 33 °C, and after 30 min 100 μL of seed solution was added.

### 2.3. Surface Modification of AuNR with MSN Shell

The synthesis of silica-coated AuNR (AuNR-TEOS) was accomplish using the Stöber method with slight modifications [26]. AuNR were placed in a 0.8 mM CTAB solution (10 mL) at a pH of 10, and the absorbance was adjusted to 1; after that, an optimization protocol was carried out to find the better synthesis conditions to obtain a homogeneous silica layer and absence of precipitation. An aliquot of 20 µL of TEOS/ethanol solution (1:4) was added 3 times every 30 min to the AuNR suspension, and the reaction was kept for 48 h at 33 °C at rest. The resulting product was centrifuged at 10,000 RPM for 15 min. To remove the CTAB, the AuNR-TEOS were suspended in 60 mL of a NH_4_NO_3_ solution in ethanol (6 mg mL^−1^) and refluxed at 60 °C for 4 h; the resulting product was named AuNR-MSN.

### 2.4. Surface Modification of AuNR with MSN Shell Functionalized with Carboxyl Groups

An optimization process was carried out to functionalize AuNR-MSN with amine groups. Different APTES volumes were added to the AuNR-MSN suspension obtained in the previous step, and conditions that showed the presence of amine signal in FTIR and absence of precipitation were selected. An aliquot of 20 µL of APTES was added to the AuNR-MSN suspended in ethanol and reaction was refluxed at 60 °C for 12 h with constant magnetic stirring; then, the product was centrifuged at 10,000 RPM for 15 min and resuspended in ethanol [22]. Subsequently, the AuNR-MSN functionalized with amino groups (AuNR-APTES) were subjected to a succinylation process using succinic anhydride. AuNR-APTES were added to 4 mL of solution of SA dissolved in DMF (40 μg mL^−1^) in the presence of triethylamine (1 mg mL^−1^). The reaction was kept for 2 h at room temperature with constant magnetic stirring [27]; the product, named AuNR-SA, was centrifuged and absorbance was adjusted to a value of 2 in water.

### 2.5. Cisplatin Encapsulation

Cisplatin was encapsulated using the AuNR-SA system by reacting cisplatin dissolved in DMSO (2 mL) at several concentrations (50, 75, 100, 250, 500 and 1000 μg mL^−1^) with 2 mL of an aqueous suspension of AuNR-SA solution. The reaction was carried out at 45 °C in the dark for 24 h under constant magnetic stirring. The product was centrifuged, and the supernatant was kept for further cisplatin quantification. The encapsulation efficiency was determined through UV-Vis spectroscopy using the OPDA colorimetric method [28].

### 2.6. Cisplatin Release Studies

AuNR-SA-cisplatin was resuspended in 5 mL of PBS at pH 7.4 or 5 and stirred for 96 h at 37 °C. To maintain cisplatin in its chlorinated form and to favor ligand exchange, PBS containing a sodium chloride concentration of 137 mM was used. Aliquots of 500 µL were taken at different times (0.5, 1, 2, 3, 4, 5, 6, 12, 24, 48, 72 and 96 h) and centrifuged, and the concentration of cisplatin in the supernatant was determined using the colorimetric method with OPDA. The original suspension was refilled with 500 µL of PBS. Cisplatin concentrations were corrected with the following formula:C_real_ = (C_measured_)(DF)(n)(1)
where C_real_ is the actual concentration, C_measured_ is the measured concentration, DF is the dilution factor (10/9) and n is the corresponding sample-taken number. Based on the correction with the above formula, the adjusted cisplatin concentration (C_real_) was obtained, from which the release percentage was calculated [28].

## 3. Results

### 3.1. UV-Vis Spectroscopy of AuNR

NH_4_NO_3_ was used to remove CTAB through ionic exchange, and UV-Vis spectroscopy measurements were performed for the samples with AuNR, AuNR-TEOS and AuNR-TEOS treated with NH_4_NO_3_. Figure 1 shows a band in ~513 nm corresponding to transverse surface resonance plasmon, while for longitudinal LPRS a band of ~752 nm is shown. AuNR-TEOS particles presented a redshift of approximately 30 nm that is associated with the change in refractive index caused by dense silica coating [25]. When AuNR-TEOS are treated with NH_4_NO_3_ to remove CTAB, a blueshift is observed in the LPRS because mesopores are formed on the surface of the silica shell, changing the refractive index [26].

### 3.2. Scanning Electron Microscopy

SEM micrographs showed the presence of nanorods with a difference in density between AuNR and silica, demonstrating the AuNR coating, corroborating the results obtained with UV-Vis (Figure 2). Through a procedure known as condensation, Si(OH)_4_ can react with the surface of AuNR; during this process, silicic acid molecules link together to form a siloxane (Si-O-Si) network, progressively giving a thin silica layer. The positively charged quaternary ammonium groups in CTAB can attract the negatively charged silicic acid molecules, facilitating the adsorption of TEOS onto the AuNR surface. This electrostatic contact accelerates the development and nucleation of the silica shell. In addition, the CTAB layer can act as a template to create the mesoporous silica shell [29,30].

The AuNR aspect ratio was determined with the average length and width, resulting in 3.2. It was observed that the length of the AuNR is in the range of 35 to 75 nm (52.4 ± 7.9 nm), and the width varies from 11 to 21 nm (15.99 ± 2.28 nm); the silica layer thickness was from 6 to 24 nm (14.74 ± 3.6 nm).

### 3.3. Surface Charge of Silica-Coated AuNR

The ζ potential is essential in understanding colloidal particle stability and behavior of nanoparticles in a liquid media; therefore, this parameter can be crucial in determining AuNR-MSN properties and future applications. Figure 3 shows values of the different stages of the AuNR-MSN synthesis; +48 mV for the AuNR­CTAB, a decrease to +10 mV is observed after treatment with NH_4_NO_3_ due to the elimination of the CTAB. For AuNR-MSN treated with SA, a reversal of the value to -21 mV is observed, which suggests that the AuNR-SA presents carboxylic acids on their surface. SA is a common coupling agent that can react with amine groups on the surface of AuNR to introduce carboxylic acid groups. From the above, it can be suggested that the functionalization treatments with amino and carboxyl groups of AuNR-MSN were successfully performed.

The ζ potential can alter the stability of AuNR by influencing the repulsion or attraction forces between suspended particles. A higher ζ potential suggests a larger repulsion force, which leads to enhanced stability and decreased aggregation. A lower ζ potential, on the other hand, can cause attraction forces resulting in instability and higher aggregation. Overall, the ζ potential is an important parameter in understanding the properties and prospective applications of AuNR-MSN, and measuring and manipulating it can provide useful insights on chemical surface modification by tracking changes in values.

### 3.4. Infrared Spectroscopy

AuNR-MSN were modified with amine groups using APTES, then with carboxylate groups using SA to promote cisplatin encapsulation. The general process is shown in Figure 1. The chemical composition of the AuNR functionalized with amine and carboxyl groups was determined by FTIR spectroscopy. Figure 4 shows a comparison of the FTIR spectra of AuNR-APTES (black) and AuNR-SA (red) to determine the presence of each ligand. The strong absorption band at 1045 cm^−1^ was associated with the asymmetric and symmetric Si-O-Si stretching, while the absorption bands around 793 and the hump at 957 cm^−1^ were assigned to the Si-O symmetric and Si-OH asymmetric vibrations, respectively [31,32]. These results confirm that the mesoporous SiO_2_ layer is successfully chemisorbed on the surface of AuNR. The AuNR-APTES spectrum shows the characteristic absorption bands of the amino groups. The broad band at 3282 cm^−1^ and another of minor intensity at 1650 cm^−1^ were associated with NH_2_ asymmetric stretch [33,34]. The peaks at 2930 and 2854 cm^−1^ correspond to the asymmetric and symmetric stretching vibrations of the methylene group, respectively, suggesting that the APTES was coupled on the surface of AuNR [35]. The spectrum of AuNR-SA shows an increase in the intensity of the bands at 1650 and 1555 cm^−1^. This can be attributed to the presence of N-H and C=O bonds of the amide group, indicating that the succinylation process was successful. The inset of Figure 4 shows the presence of a new absorption band at 1708 cm^−1^ which can be ascribed to the C=O stretching vibration of the carboxyl group, indicating that the acid anhydride was hydrolyzed to carboxylic acid [12,36].

### 3.5. Encapsulation of Cisplatin

The amount and percentage of encapsulated cisplatin was determined from the supernatant by derivatization with OPDA (Figure 5). The amount of drug loading was relatively high for AuNR-MSN that were subjected to high concentrations of initial cisplatin. However, the percentage of encapsulated cisplatin decreased from 74% to 31% as cisplatin concentration increased. The carboxylate groups on the surface of the AuNR can interact with cisplatin through coordination chemistry, forming a surface-bound complex [37,38]. This fact can enhance the drug delivery properties of the nanoparticles, improving the solubility and stability of cisplatin, making it more available for cellular uptake and interaction with target molecules [39]. The oxygen-containing carboxylic acid on the nanoparticles surface can serve as a donor of atoms to create coordination bonds with the metal platinum ion in the cisplatin molecule. The complex can also protect cisplatin from degradation in the body and help to target the drug to the site of action [37,38].

According to the physical stabilization presented by the samples, it was observed that the nanosystem presented resuspension issues, which may be due to cisplatin covering the surface of the AuNR-SA completely at high concentrations. The best resuspension conditions were observed in the cisplatin concentration range of 50 to 75 µg/mL. The formation of the surface-bound complex can affect the properties of the nanoparticles’ surface, such as its surface charge, wettability, and solubility. The 75 µg/mL concentration was chosen for further characterization since the nanoparticles subjected to it presented good resuspension conditions, the highest possible encapsulation percentage (58%) and a considerable mass of encapsulated cisplatin (83.93 µg).

### 3.6. Cisplatin Release Kinetics

Cisplatin release from AuNR-SA was evaluated under simulated physiological conditions (37 °C, pH 7.4) using phosphate-buffered saline with a chloride concentration of 137 mM. During the first 6 h a rapid release was observed, which can be attributed to the cisplatin absorbed on the surface of the AuNR-SA. After the first 12 h, a slow release was observed until reaching 60% of the encapsulated drug in a time of 96 h (Figure 6). In a physiological pH environment, the release of cisplatin from carboxylic acid surface-functionalized nanoparticles may be slower compared to an acidic environment. This behavior is probably because carboxylic acid groups on the surface of the nanoparticles are typically deprotonated at this pH. Acidic pH showed a faster release of 72% compared to 51% at pH 7.4 after 96 h; in this case the carboxylate acid groups on the nanoparticle surface become protonated, which could cause a decrease in the stability of the complex. This can lead to the release of cisplatin from nanoparticles [40].

Tan et al. [41] studied the surface modification of cisplatin-complexed AuNPs and its effect on their colloidal stability, drug loading and drug release via a carboxyl-terminated dendron ligand like the present study but using a different carboxylate ligand. They investigated in vitro release under acidic and neutral pH settings, finding that under acidic conditions, a release burst occurred within two days, followed by a gradual release between two and ten days, resulting in a tenfold rise in Pt^II^. The sustained release of cisplatin is of great importance for drug delivery in cancer chemotherapy to avoid dissipation and inactivation of the drug in the physiological environment before arrival at the tumor site of interest. It is well-known that adsorption does not ensure sustained drug release, so encapsulation using a coordination bond is being evaluated [13]. In these cases, the slow and controlled release of cisplatin from nanoparticles in a neutral pH can help minimize systemic toxicity and improve the therapeutic drug efficacy. Nejad and Urbassek [42] reported a molecular dynamic study to describe diffusion of cisplatin through polar silica pores; they found that cisplatin motion in pores is governed by short adsorption and desorption events. In the present study, we hypothesized that cisplatin and carboxylates form a coordination complex; the first step in diffusion probably is the breakdown of that complex following the events of adsorption–desorption until cisplatin is completely released from the nanoparticle.

In silico approaches are recognized to be excellent tools for researching biological and clinical processes, and hence can considerably improve the process of generating new nanoparticles-based medications and refining existing ones. This knowledge can be utilized to create more effective and targeted drug delivery systems, enhance the safety of AuNP-based therapies and guide the development of new nanomedicines. Aborig et al. [43] developed a model that tends to over-predict AuNP amounts in organs with very low or near-zero exposure (heart, kidney and lung) and under-predict amounts in portal organs (spleen and stomach) after intraperitoneal administration, achieving an absolute average error of 1.48 compared to in vivo data. The simulations can help us predict where cisplatin can accumulate in each tissue and at what specific time by mathematical models; this can potentiate phototherapy and chemotherapy synergism. For example, Lazebnik et al. [44] created a network of interconnected nodes to represent the different components of a biological system for analyzing the relationships between red blood cell (RBC) coated polymer nanoparticles and micechlorotoxin-conjugated iron oxide nanoparticles biodistribution, obtaining a fitting of 0.84 ± 0.01 and 0.66 ± 0.01 (mean ± standard deviation), respectively, comparing the in vivo values and the in silico results. Having a more accurate predictor based on simulated conditions will help create therapies that specifically target tissues or organs more easily and correlate biodistribution with a drug dissolution profile before in vivo tests.

## 4. Conclusions

In this study we incorporated cisplatin in carboxylate-group-modified mesoporous silica-coated AuNR by forming a coordination complex. The presence of carboxylate groups on the surface of the AuNR-SA permitted cisplatin incorporation through the creation of coordination bonds with the platinum ion of the cisplatin molecule. The complex formation can protect the drug from degradation and helps to target it to the site of action for a controlled release. Cisplatin release profiles showed a pH-dependent drug release behavior; after 96 h we observed a drug release of 72% and 57% for pH 5.5 and 7.4, respectively. These results show the potential of the AuNR-SA-cisplatin system as a possible effective chemo-photothermal agent to maximize therapeutic efficacy and minimize dose-dependent side effects against cancer diseases. This system still needs to be evaluated in terms of the synergistic effect of drug delivery and photothermal therapy against cancer cell lines. Cisplatin is frequently used in conjunction with other medications to improve efficacy or overcome drug resistance. Future research should look into the possibility of combining cisplatin-loaded AuNR with other pharmaceuticals to improve cancer treatment outcomes, such as immunomodulators or targeted therapies. AuNR-MSN have demonstrated exceptional potential as a platform for various medication administration since they can improve efficacy while minimizing toxicity. However, there are several limitations to this strategy that should be considered. While the MSN can improve cisplatin selectivity for cancer cells, there is still a risk of off-target effects and damage to healthy cells. More research is needed to improve these platforms’ targeting specificity to limit the possibility of off-target effects while exploring targeted therapies.

## Data Availability

Data sharing not applicable.

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
