# Peer review of "Gold Nanorods with Mesoporous Silica Shell: A Promising Platform for Cisplatin Delivery"

_micromachines, 2023, doi:10.3390/mi14051031_

Round 1
Reviewer 1 Report
In the manuscript titled Gold Nanorods with Mesoporous Silica Shell: A Promising Platform for Cisplatin Delivery, Quiñones et al. report the development of silica coated GNRs able to load cisplatin and to release it in aqueous environment. It is my opinion that the present work seriously lacks of novelty and scientific soundness, since the reported approach appears as a patchwork of different approaches already well discussed, described and reported in the literature. In fact, the coating of GNRs with silica, the modification of silica with APTES + succinic anhydride and the coordination of cisplatin to carboxylated silica surface s has been reported many times in the last 10-15 years. In order for this work to minimally increase its significance, I recommend the authors to perform additional studies to complete this description. For example, it would be interesting to see release tests in vitro, to assess cytotoxicity of the released cisplatin towards cancer cells. Additionally, the photothermal properties of the final nanosystem should be evaluated, and also the assessment of whether the increase of temperature caused by the photothermal effect may lead to an increase in the rate of cisplatin release. Also, diagnostic potential of the nanoplatform should be evaluated. For this reasons, I believe that with any of the listed additional experiment and only limiting to in silico studies, this work represent a very preliminary investigation, not suitable for publication on Micromachines.
A few additional comments:
1) In row 60, it is worth citing literature papers where it is described the combined use of photothermal therapy and chemotherapy such as in DOI: 10.2147/IJN.S197265.
2) The surface plasmon resonance of gold nanorods is often exploited to achieve contemporary tumor diagnosis and therapy with a single theranostic platform, such as in DOI: 10.1016/j.pacs.2022.100400. This type of approach deserves to be treated in an independent paragraph of the introduction section.
3) In Materials and Methods, paragraph 2.3, in the experimental conditions for the silica coating of GNRs the authors express the concentration of gold nanorods in terms of their absorbance, which was adjusted to 1. This might be trivial, since different batches of synthetized GNRs give rise to different size distributions and therefore slightly different spectral profiles, that may be larger and less intense for polydisperse batches or thinner and more intense for monodispersed batches. To make this point more clear and therefore the whole procedure more reproducible, I recommend to authors to express GNRs concentration as atomic gold concentration (mM).
4) In Figure 2, it is impossible to read to which length the scale bar corresponds.
5) The manuscript lacks of a main scheme reporting the different steps of the surface modification of gold nanorods, reporting the chemical bonds that are formed between APTES and the silica shell, between succinic anhydride and APTES, and from surface-conjugated succinic acid and cisplatin.
Minor editing of English language required
Reviewer 2 Report
This research focus on nanoparticle drug delivery systems using gold nanoparticles covered in biological layer. The authors show their design is useful for long-term delivery (96 hours) which is much better than previous attempts. Overall, the idea and results are interesting and my main concerns are the writing of the paper which is a bit short and limited for the non-expert reader and can make it hard to use and reproduce the proposed study, limiting its usefulness. As such, I suggest the authors improve the following points before the paper can be accepted for publication:
1. The related work part from the introduction is too limited. The authors must review in more detail other gold-based nanoparticles used as part of a drug delivery. For example: E Ben-Akiva, R. A. Meyer, H. Yu, J. T. Smith, D. M. Pardoll, and J. J. Green. “Biomimetic anisotropic polymeric nanoparticles coated with red blood cell membranes for enhanced circulation and toxin removal”. In: Science Advances 6.16 (2020). An older example: R. Popovtzer, A. Agrawal, N. A. Kotov, A. Popovtzer, J. Balter, T. E. Carey, and R. Kopelman. “Targeted Gold Nanoparticles Enable Molecular CT Imaging of Cancer”. In: Nano Letters 8 (2008).
2. Following the previous comment, while the authors are biologically-oriented, it is impossible to ignore the growing field of nanoparticles simulation. Thus, I strongly recommended the author to include, even shortly, a review of this approach and how it is interact with their research and possible research in their line of work. The authors can include the following, along side others, works:
A. https://doi.org/10.1101/2022.07.13.499855
B. https://pubmed.ncbi.nlm.nih.gov/31013763
C. https://doi.org/10.1101/2022.07.12.499805
3. While section 2 is well-written, it seems to me like a set of random decisions. I think it can be useful if the authors could motivate their results. It is fine to say "we tried xxx and yyy and learned that zzz is good idea cause ----". It can help others to learn the thought process required to get a good nanoparticle solution.
4. We see many mean +- STD values. Please state the number of samples somewhere.
5. Fig 3 & 6 - explain the error bars in the caption.
6. The summary section is quite limited as well. Please add possible future work. Please explain the limitations of the current research. How one can use the shown results in practice?
There are several typos and grammatical errors across the manuscript. However, the main issue is the very-long sentences that are a bit confusing. I suggest the authors try to write in short and clear sentences - this will make it easier to be clear what they try to convey in each statement.
Round 2
Reviewer 1 Report
The main issues have been addressed by the authors in the revised version.
Minor editing of English language required
Author Response
Response to reviewer 1 (Second Round)
Minor editing of English language required
We have worked on the quality of English in the manuscript.
Reviewer 2 Report
The authors did some modifications to the manuscript, improving it a lot. However, I feel there are several points that are not clear, and after addressing them, the manuscript can be published. First, it seems the authors mostly ignore the subject of nanoparticle simulation as drug delivery agents. While the added text is fine in general it ignores the main subject. I can understand the authors do not want to diverge too much from their subject but in silico experiments go hand in hand nowadays with in vivo and in vitro. Thus, I encourage again the authors to include, even shortly the works suggested in the previous round and many others. The current reference list is 44 references, this is quite short for a biological paper and it just add to the work's volume and robustness. In addition, the last section still does not provide an applicative usage of this work. Maybe the authors can provide an example or two. Lately, the authors much explain in more details the graphs in the captions, it is really hard to follow in the current form.
I did not notice much change from the previous version.
